# A New Method of Priority Assignment for Real-Time Flows in the WirelessHART Network by the TDMA Protocol

**DOI:** 10.3390/s18124242

**Published:** 2018-12-03

**Authors:** Yulong Wu, Weizhe Zhang, Hui He, Yawei Liu

**Affiliations:** 1School of Computer Science and Technology, Harbin Institute of Technology, Harbin 150001, China; yulongwu@hit.edu.cn (Y.W.); hehui@hit.edu.cn (H.H.); liuyawei@hit.edu.cn (Y.L.); 2Cyberspace Security Research Center, Pengcheng Laboratory, Shenzhen 518055, China

**Keywords:** WirelessHART network, delay analysis, real-time systems, multi-channel processing, simulation modeling

## Abstract

WirelessHART is a wireless sensor network that is widely used in real-time demand analyses. A key challenge faced by WirelessHART is to ensure the character of real-time data transmission in the network. Identifying a priority assignment strategy that reduces the delay in flow transmission is crucial in ensuring real-time network performance and the schedulability of real-time network flows. We study the priority assignment of real-time flows in WirelessHART on the basis of the multi-channel time division multiple access (TDMA) protocol to reduce the delay and improve the ratio of scheduled. We provide three kinds of methods: (1) worst fit, (2) best fit, and (3) first fit and choose the most suitable one, namely the worst-fit method for allocating flows to each channel. More importantly, we propose two heuristic algorithms—a priority assignment algorithm based on the greedy strategy for *C* (WF-C) and a priority assignment algorithm based on the greedy strategy for *U*(WF-U)—for assigning priorities to the flows in each channel, whose time complexity is $O(max\left(N*m*log\left(m\right),{\left(N-m\right)}^{2}\right))$. We then build a new simulation model to simulate the transmission of real-time flows in WirelessHART. Finally, we compare our two algorithms with WF-D and HLS algorithms in terms of the average value of the total end-to-end delay of flow sets, the ratio of schedulable flow sets, and the calculation time of the schedulability analysis. The optimal algorithm WF-C reduces the delay by up to 44.18% and increases the schedulability ratio by up to 70.7%, and it reduces the calculation time compared with the HLS algorithm.

## 1. Introduction

With the introduction of the Industry 4.0 concept, IoT devices face many challenges such as sustainability and security, among which one of the most important is real-time performance [1,2]. In this paper, we propose two heuristic algorithms to assign priority to real-time flows in WirelessHART networks, and focus on the real-time performance of these two algorithms. The WirelessHART network is widely used in many fields, such as industrial manufacturing, automation control, and information collection. WirelessHART includes HART, EDDL, and IEEE 802.15.4 protocols. The TDMA protocol is widely used in wireless sensor networks, and it consists of the following devices [3].
*Field devices* are nodes in the WirelessHART network that can be regarded as source, transmission, or destination nodes. These nodes can collect, transmit, or receive information if they function as source, route, or destination nodes, respectively.*Gateway* limits the range of the WirelessHART network and allocates an IP address to every node to ensure that every node’s IP is unique. Hence, the source node that sends the information in the network can be determined. Backtracking the information through the gateway is convenient and accurate. The WirelessHART network has only one gateway.The *network manager* is a global administrative unit for the WirelessHART network. It contains all of the topological structure information of the network. When a new node wishes to join the network, it has to obtain its own IP address from the gateway through the network manager. The information of this new node, such as its neighbor nodes, is saved at the host by the network manager.

Among the many protocols of the WirelessHART network, TDMA is to the most suitable for the real-time feature, the veracity of dates, and the power requirement of the WirelessHART network [4]. In general, we consider the flows to be periodic; i.e., all flows send packets again after a certain interval. Similarly, we can think of an aperiodic real-time flow as a real-time flow with an infinite length period. Hence, the TDMA protocol is also suitable for acyclic real-time flows. The TDMA protocol allows the time to be divided into several periodic frames, and each frame is divided into an equal numbers of slots. To guarantee the real-time characteristic, a device is only allowed to send and receive data in a specified slot. The TDMA protocol allows a source node to send many of the same packages to the destination node through different routes. This guarantees the accuracy of data transmission by increasing redundancy. Hence, we think of these same packets in different routes as separate flows in this study. In other words, if a source node sends the same packets through *k* routes, then we think of them as *k*-independent flows with the same period and deadline. The TDMA protocol also allows nodes that do not send or receive data to enter sleep mode and meet the power requirement of the WirelessHART network.

**Real-time flow:** The information collected by the source node in the WirelessHART network is transmitted as a packet. We define a real-time flow (referred to as flow) as a packet sent from the source node to the destination node. The most obvious difference between real-time flow and regular flow is that real-time flow has strong real time. The real-time flow must be finished before its deadline. Otherwise, there will be serious consequences, such as a system crash.

**Priority assignment of flow:** We consider a set of *N* flows, $F=\left\{F_{1},F_{2},\ldots ,F_{N}\right\}$, and a priority assignment strategy $f=\left\{1,2,\ldots ,N\right\}$. The number of priorities is equal to the number of flows, and the priorities are in a strict decreasing order, i.e., $f_{1}>f_{2}>\ldots >f_{N}$. Hence, no two flows have the same priority. Specifically, the priority of flow $F_{j}$ is higher than that of flow $F_{i}$ if and only if j<i. We consider flow $F_{i}$ with priority *j*. If the condition $\forall F_{i}\in F,\exists j\in f,s.t.f:F_{i}=j$ is met. Furthermore, we say a priority assignment strategy *f* is satiable if the flows in the set all meet their deadline.

**Optimal priority assignment:** We consider a schedulability test *S* and a priority assignment algorithm *A*. If is a priority assignment policy is available for a flow set that is satisfied in schedulability test *S*, *A* can also provide a satisfactory priority assignment. We then call *A* an optimal priority assignment. In other words, if a flow set cannot be satisfied by using *A* assignment priority in the schedulability test *S*, then no assignment algorithm can satisfy the flow set in schedulability test *S*.

The contributions of our work are as follows: (1) We transform the multi-channel flow model of the WirelessHART network into a multi-processor real-time model of the CPU. We allocate the flows to channels by using the worst fit algorithm. (2) We show that the problem of priority assignment in a single channel makes the set of flows acceptable, which is an NP-complete (NPC) problem. Then we propose two heuristic algorithms, namely, a priority assignment algorithm based on the greedy strategy for *C* (WF-C) and a priority assignment algorithm based on the greedy strategy for *U*(WF-U). (3) We build a new WirlessHART network simulation model according to the actual environment. (4) We compare our two algorithms with WF-D and HLS algorithms in terms of the average value of the total end-to-end delay of flow sets, ratio of schedulable flow sets, and calculation time of the schedulability analysis.

## 2. Background

We consider the WirelessHART network model with *m* channels based on the time division multiple access (TDMA) protocol. Each channel is divided into several periodical frames, and each frame is divided into slots with the same number. The length of every slot is 10 ms [5]. All information in the WirelessHART network is transmitted as packets. We call the packet that is sent from the source node to the destination node a flow.

Each node in the WirelessHART network can be viewed as a source, middle, or destination node [6]. Every node in a slot can receive or send a packet. That is, flow $F_{i}$ sent from node *a* to node *b* costs one slot, and node *b* receiving it and sending it to node *c* also costs one slot. Each node can only perform one type of function (either sending or receiving a packet) in one slot.

To ensure transmission accuracy, we allow a packet to be sent from the source node to the destination node in different routes. The advantage of this policy is that can avoid blocking by the disabled node, bad routes, or blacklisting. For example, we assume that a packet will be set from node *C* to node *D* in Figure 1a. We send the same packet by two routes: (1) C→H→I→D and (2) C→D. This packet cannot be transferred through Route (1) if the node *I* belongs to the blacklisting. However, this packet can still be transferred through Route (2). Hence, we guarantee that this packet can be received by node *D* by adding a route. In other words, this policy guarantees the accuracy of data transmission by increasing redundancy.

## 3. Real-Time Flow Model

We consider the set *F* with *N* flows mentioned above. A flow $F_{i},F_{i}\in F$ consists of three parameters $\langle C_{i},T_{i},D_{i}\rangle$. That is, for a flow $F_{i}$, $C_{i}$ is its execution time that contains the transmission time, $T_{i}$ is its minimum interval time between two consecutive jobs, and $D_{i}$ is its relative deadline. We also consider that the real-time flow model is deadline-constrained, i.e., D≤T.

The utilization of flow $F_{i}$ is denoted by $U_{i}=C_{i}/T_{i}$. We let $r_{i}$ and $a_{i}$ be the release and arrival time of $F_{i}$, respectively. The $r_{i}$ is the instant that flow $F_{i}$ sends its packet from the source node. $a_{i}$ is the instant that the packet arrives at the destination node. Hence, the real transmission time of $F_{i}$ is $t_{i}=a_{i}-r_{i}+1$. We denote the end-to-end delay as $delay=C_{i}+t_{i}$ shown in Figure 2. The worst-case end-to-end delay (WETED) of $F_{i}$ as $R_{i}$.

The multi-channel real-time flow model of the WirelessHART network can be transformed into a multi-processor real-time model of a CPU [7,8]. On the basis of the model described above, *m* channels can be transformed into *m* processors in the multi-processor model. *n* flows can be transformed into *n* real-time tasks, and each task of the multi-processor model also has three parameters $\langle C_{i},T_{i},D_{i}\rangle$. WETED can be transformed into the worst-case response time (WCRT) of the multi-processor model.

## 4. The End-to-End Delay Analyze

In the WirelessHART network, the end-to-end delay is the time when a job of flow $F_{i}$ is sent from the source node to the destination node to receive and execute the carried packet. In this paper, the slot is the smallest unit of time, which means that high-priority flows can no longer preempt low-priority flows in one slot. A flow with high priority must wait until flows with low priority finish this slot.

Flow $F_{i}$ is blocked in two ways, namely, (1) channel contention and (2) transmission conflicts, by flows with higher priorities.

### 4.1. Channel Contention

We consider a real-time flow model with *m* channels. When a free channel is present, the newly released job of a flow is sent to the channel for execution. When no channel is idle, the newly released high-priority job of a flow preempts the channel that transmits the low-priority job of the flow in the beginning of the next slot. We use $\Omega_{i}\left(t\right)$ to represent the total channel competition caused by $F_{i}$ for all the jobs of flows with a priority that is higher than that of flow $F_{i}$ in time *t*.

For flow $F_{i}$, $\Omega_{i}\left(t\right)$ cannot be accurately calculated if we use a global priority assignment strategy in the multi-channel model. Considering that we do not know when the level-*k* busy period [9] starts, we use the $\Omega_{i}\left(t\right)$ upper bound [10,11,12] to calculate the upper bound of the WETED and set it to $R_{i}$.

### 4.2. Transmission Conflicts

In the WirelessHART network, when the topology is determined, the transmission conflict between two flows with overlapping transmission routes is a constant [7]. For example, two flows are shown in Figure 1a, $F_{i}$ and $F_{j}$. We assume that $F_{j}$ has higher priority than $F_{i}$, i.e., j<i. Flow $F_{i}$ sends a packet from node *A*, through nodes *C*, *D*, and *F*, to node *G*. Flow $F_{i}$ sends a packet from node *B*, through nodes *C*, *H*, and *I*, to node *D*. The common nodes of two flows are *C* and *D*. There are two cases of transmission conflicts: (1) full transmission conflicts and (2) half transmission conflicts. The two cases are going to cost two slots and one slot, respectively.

The example of the first case is shown in Figure 1a with node *C*. We assume that $F_{i}$ and $F_{j}$ arrive at node *C* at the same time. The transmission operation of node *C* is shown in Figure 1b. We assume that $F_{i}$ and $F_{j}$ arrive at Time Slot 1. Node *C* first costs two slots to receive the packet of flow $F_{j}$ from node *A* and sends it from node *C* to node *D*, respectively. $F_{i}$ must wait for two slots because the priority of $F_{i}$ is lower than that of $F_{j}$. The packet of flow $F_{j}$ then starts to be processed. Hence, the first case of transmission conflicts of $F_{i}$ is blocked for two slots by $F_{j}$ at node *C*.

The example of the second case is shown in Figure 1a with node *D*. Like the first case, we also assume that $F_{i}$ and $F_{j}$ arrive at node *D* at the same time. The transmission operation of node *D* is shown in Figure 1c. We assume that $F_{i}$ and $F_{j}$ arrive at Time Slot 1. Different form the first case, $F_{i}$ only needs to wait for one slot because flow $F_{j}$ ends at node *D*, i.e., node *D* do not need cost one slot to send the packet of $F_{j}$ to other nodes. Hence, the second case of transmission conflicts of $F_{i}$ is blocked for one slot by $F_{j}$ at node *D*.

Therefore, $F_{i}$ is blocked for three slots by $F_{j}$ in this topology structure in the worst case. We define $\Delta \left(i,j\right)$ as the transmission conflicts caused by $F_{j}$ to $F_{i}$, where $F_{j}$ has a higher priority than $F_{i}$.

### 4.3. Schedulability Test

If we know the priority assignment for the flow set, we can determine whether the flow set is schedulable or not by calculating the end-to-end delay in the worst case [8]. We can obtain the end-to-end delay in the worst case by calculating the minimum value of *y* in Equation (Equation 1), where $t^{*}$ is the minimum *t* in Equation (2).

If one of a flow’s *t* or *y* is greater than its deadline, we call it unschedulable, or schedulable at that priority. If all the flows in the set are schedulable, we say that this priority assignment policy is satiable.
(1)$$\begin{array}{c} y=t^{*}+\sum_{j\in hp(i)}\left\lceil \frac{y}{T_{j}}\right\rceil \;\Delta \left(i,j\right) \end{array}$$
(2)$$\begin{array}{c} t=\left\lfloor \frac{\Omega_{i}\left(t\right)}{m}\right\rfloor +C_{i}. \end{array}$$

## 5. The Priority Assignment of Real-Time Flows Based on the Worst-Fit

In this section, we first introduce several algorithms (worst-fit, best-fit, and first-fit) to allocate the flows and explain why we use the worst-fit. We then introduce several priority assignment algorithms based on the worst-fit.

We think of the problem of allocating the flows to channels as a bin packing problem, since each channel is equivalent to a processor and the utilization upper bound of the processor is 1. Hence, we assume the channel as a bin with a capacity of 1. Each flow is viewed as cargo whose volume equals its utilization. There are three state-of-the-art existing allocation methods: (1) worst-fit, (2) best-fit, and (3) first-fit. The descriptions and characters of them are as follows:Worst-fit is to allocate the flow to the channel with the maximum remaining capacity of all channels each time. This method can make the use part of each channel more even.Best-fit is to allocate the flow to the channel with the minimum remaining capacity of all channels each time. This method can reduce the remaining resources of each channel as much as possible.First-fit is to allocate the flow to the first channel, and the remaining utilization of this channel is not less than the utilization of this flow. This method can reduce search time.

To make efficient use of each channel, our aim is to distribute the flow to each channel as evenly as possible, while ensuring that the capacity of each channel is within the allowed range. Hence, the worst-fit method is the best choice.

The multi-channel fixed priority transmission scheduling problem for WirelessHART networks can be mapped to the CPU scheduling on a global multiprocessor platform [8] by Saifullah et al. Similarly, we regard each channel for multiple channels WirelessHART networks as a homogeneous and independent processor for CPU scheduling; i.e., each channel is mapped to a processor and there is no interference between any two processors. We then distribute the flows to the channels, i.e., the flow can be transmitted in a channel if and only if it has been distributed in this channel. At last we assign priority to the set of flows in each channel.

This method avoids the analysis of channel contention and uses a new approach to analyze the WETED accurately in Equation (Equation 3), where hp(i) is a set of flows that have a higher priority than $F_{i}$. $R_{i}$ can be calculated with $C_{i}$ as the starting value.
(3)$$\begin{array}{c} R_{i}=C_{i}+\sum_{j\in hp(i)}\left\lceil \frac{y}{T_{j}}\right\rceil \;\left(C_{j}+\Delta \left(i,j\right)\right). \end{array}$$

### 5.1. Method of Allocating Flows to Channels based on the Worst Fit

We assume that a set of *N* flows $F=\left\{F_{1},F_{2},\ldots ,F_{N}\right\}$ exists in the WirelessHART network with *m* channels. We denote $U_{sum}^{k}$ as the total of the utilization of the *k*-th channel, and it can be calculated with Equation (Equation 4). Similarly, we denote $U_{sum}$ as the total utilization of the set, which can be calculated with Equation (5).
(4)$$\begin{array}{c} U_{sum}^{k}=\sum_{F_{i}\in channel\;k}U_{i}. \end{array}$$
(5)$$\begin{array}{c} U_{sum}=\sum_{k=1}^{m}U_{sum}^{k}. \end{array}$$

A necessary condition for flow set schedulability on a multi-processor platform is $U_{sum}\leq m$, where *m* is the number of processors. This condition means that the set can never be scheduled if $U_{sum}>m$. We also determine that the set can never be scheduled for a single channel if $U_{sum}^{k}>1$. We define several symbols to describe the worst-fit method as follows:
$Re^{k}:\;$ the current remaining utilization of the *k*-th channel;$Ch^{k}:\;$ the set of flows that belong to Channel *k*;$UCh^{k}:\;$ the set of flows that have not been assigned priority in the *k*-th channel;$ACh^{k}:\;$ the set of flows that have already been assigned priority in the *k*-th channel;ReF: the set of flows that have not been allocated a channel;$C_{i}^{j}:\;$ the execution time of flows $F_{j}$ at priority *i*;$R_{i}^{j}:\;$ the WETED of flows $F_{j}$ at priority *i*.

Before the worst-fit algorithm begins, no flows are allocated to the channel. All flows are stored in the set ReF. The flow in the set ReF is continuously allocated to the channel with the implementation of the worst-fit algorithm. It terminates until all flows in ReF are allocated or until no channel can accept the current allocated flow.

The steps to allocate flows to the channels are as follows:Initialize $Re^{k}=1$, $Ch^{k}=\emptyset$, where k∈m, and then proceed to Step 2.If set ReF is empty, then the priority assignment ends. Return *True* and the algorithm stops. Otherwise, jump to Step 3.Sort the remaining utilization in a descending order for each channel, and then jump to Step 4.Allocate one flow $F_{i}$ from set ReF into Channel 1. If $Re^{1}-U_{i}<0$, then we denote the flow set as unschedulable and return *False*. Otherwise, $Re^{1}=Re^{1}-U_{i}$. Delete $F_{i}$ from ReF and jump to Step 2.

Specifically, no channel can accommodate flow $F_{i}$ if Channel 1 cannot accommodate it because the channels have already been sorted in Step (2). We present a pseudo-code of the worst-fit method as Algorithm 1.

**Algorithm 1** The worst-fit algorithm.**Input:** The set of flows F=F1,F2,…,FN**Output:**
**bool** WF  1: **function**
Worst-Fit  2:    $Re^{k}\leftarrow 1,\;where\;k=\left\{1,2,\ldots ,m\right\}$  3:    $Ch^{k}\leftarrow \emptyset$,$\;where\;k=\left\{1,2,\ldots ,m\right\}$  4:    **for**
i=1ton
**do**  5:        sort Re in descending order  6:        **if**
$Re^{1}-U_{i}<0$
**then**  7:           **return**
*False*  8:        **else**  9:           Ch1=Ch1⋃Fi10:           ReF=ReF−Fi11:        **end if**12:    **end for**13:    **return**
*True*14: **end function**

We can assign the priority of flows independently in the single channel if the worst-fit method returns True in Algorithm 1.

We then consider the priority assignment in a single channel. As shown in Theorem 1, the problem of priority assignment in a single channel (PAS) makes the set of flows acceptable, which is an NPC problem. Hence, we propose two heuristic algorithms: a priority assignment algorithm based on the greedy strategy for *C* (WF-C) and a priority assignment algorithm based on the greedy strategy for *U*(WF-U). Both are based on an analysis of Equation (Equation 3).

**Theorem** **1.**
*The PAS problem is an NPC problem.*


**Proof** **of** **Theorem** **1.**First, we prove that the PAS problem is an NP problem. If we assign a priority assignment strategy for a set of flows, we can easily determine the delay of each flow in polynomial time according to Equation (Equation 3). By summing the delay of each flow, we can verify whether the total delay can be satisfied or not under the priority assignment strategy. Thus, the PAS problem is an NP problem.Second, we prove that PAS is an NPC problem. During the maximum continuous working time [13], if all flows have only one job instance, i.e., the inequality in Equation (Equation 6) is satisfied, where $T_{i}$ is the period of $F_{i},F_{i}\in F$, then Equation (7) can be used to calculate the delay of flow $F_{i}$. Therefore, solving the PAS problem can be transformed into solving the minimum delay of the lowest priority (DLP) problem.
(6)$$\begin{array}{c} T_{i}\leq \sum_{j\in hp(i)}\left[C_{j}+\Delta \left(i,j\right)\right]. \end{array}$$
(7)$$\begin{array}{c} R_{i}=\sum_{j\in hp(i)}\left(C_{j}+\Delta \left(i,j\right)\right). \end{array}$$We know that the traveling salesman problem (TSP) is an NPC problem [14]. We stipulate that *n* cities in the TSP problem correspond to *n* flows in the DLP problem. We define $\Delta \left(i,j\right)+C_{j}$ as the cost from city *i* to city *j*. We define the path from the starting city in the TSP problem as the sequence from low priority to high priority in the DLP problem. Assuming that the starting city of the TSP problem is *s*, the last arriving city is *l*, and the solution is ST, then the solution of the DLP problem is ST−Δ(l,s). In summary, the TSP problem is reduced to a PAS problem. Thus, the PAS problem is an NPC problem. □

### 5.2. WF-C Algorithm and WF-U Algorithm

Through an analysis of Equation (Equation 3), we find that one of the most important factors that influence WETED is the execution time of flow $F_{j}$ with a higher priority than flow $F_{i}$. Thus, we start by assigning the lowest priority to the flow set. According to the greedy strategy, the current priority is assigned to the flow in $UCh^{k}$ that has the maximum execution time in the schedulable flows. This algorithm can avoid high priority flows with a long execution time causing too many blocks to low priority flows and missing their deadline. The steps of the WF-C algorithm are as follows:Initialize $ACh^{k}=\emptyset$ and $UCh^{k}=Ch^{k}$ and then jump to Step 2.The priority is assigned from the lowest priority, and if set $UCh^{k}$ is empty, then the priority assignment policy is returned and the algorithm ends. Otherwise, jump to Step 3.Calculate the WETED of all flows in $UCh^{k}$ at the current priority. If no flow meets its deadline, then the algorithm returns *False* and the algorithm ends. Otherwise, jump to Step 4.Assign the current priority to the flow in Step 2 that meets its deadline and has the longest execution time. Then raise the current priority and jump to Step 2.

The pseudo-code of the WF-C method is shown in Algorithm 2.

**Algorithm 2** The WF-C algorithm.**Input:**Chk**Output:** the scheme of priority *f*  1: **function**
WF-C  2:    ACHk←∅  3:    UCHk←ACHk  4:    **for**
i=length(CHk)downto1
**do**  5:        Calculate all Rij,wherej∈UChk  6:        **if** all Rij>Dj,wherej∈UChk
**then**  7:           **return**
*False*  8:        **else**  9:           Fj←i // where j=maxCij10:        **end if**11:    **end for**12:    **return**
*f*13: **end function**

The WF-U algorithm is similar to the WF-C algorithm. The difference between these two algorithms is the condition for determining priority of flows. The first three steps of the WF-U algorithm are the same as those of the WF-C algorithm in Section 5.2, and the longest execution time is replaced with the maximum utilization in Step 4.

The pseudo-code of the WF-U method is shown in Algorithm 3.

**Algorithm 3** The WF-U algorithm.**Input:**Chk**Output:** the scheme of priority *f*  1: **function**
WF-U  2:    ACHk←∅  3:    UCHk←ACHk  4:    **for**
i=length(CHk)downto1
**do**  5:        Calculate all Rij,wherej∈UChk  6:        **if** all Rij>Dj,wherej∈UChk
**then**  7:           **return**
*False*  8:        **else**  9:           Fj←i // where j=maxUij10:        **end if**11:    **end for**12:    **return**
*f*13: **end function**

### 5.3. Time Complexity

In this section, we discuss the time complexity of the WF-C and WF-U algorithms. According to Algorithms 2 and 3, we can find that the calculation steps of WF-C and WF-U are basically the same, except that the condition for determining priorities are different. Hence, we just discuss the time complexity of WF-C.

We assume there are *m* channels and a flow set with *N* flows. According to Algorithm 2, we know that WF-C contains two parts, which are to allocate flows to channels and to assign priority to flows. The time complexity of the first part, O(N∗m∗log(m)), is equivalent to the time complexity of ordering N times for m processors (as is known, the time complexity of the quicksort of *m* elements is m∗log(m)).

According to Equation (Equation 3), we know that the number of calculations for a certain flow $F_{i}$ depends on the number of elements in hp(i). For example, we assume that |hp(i)|=k. We can then calculate that the number of calculations for $F_{i}$ is 3k (the 3 indicates three basic operations, the plus at the end of Equation (Equation 3), the division, and the ceil). There are at most N−m+1 flows in a channel by using the worst-fit algorithm. We can derive that the time complexity of the second part is $O({\left(N-m\right)}^{2})$ from Algorithm 1.

From what has been discussed above, the time complexity of WF-C is O(X), where *X* is $\max(N*m*\log\left(m\right),{\left(N-m\right)}^{2})$. Moreover, the WF-U has the same time complexity as WF-C.

## 6. Simulation Experiment

In this section, we propose a new simulation model based on the reality, and conduct many comparative experiments on the four algorithms, namely, a heuristic search (HLS) algorithm (which adds two discriminant conditions to find a feasible solution based on changing priorities of two tasks locally) [7], the worst case fit strategy for the deadline monotonic (WF-D) algorithm (which sends flows into channels based on Algorithm 1, and then assigns priority by DM strategy), the WF-C algorithm, and the WF-U algorithm.

We then compare the performance of these algorithms in terms of the average value of the total end-to-end delay of flow sets, the ratio of schedulable flow sets, and the calculation time of the schedulability analysis according to the experimental results.

On the basis of the actual environment [15,16,17], we define several symbols to describe our simulation model in Table 1. We assume that the number of flows in each flow set is represented by *N*. The number of channels in the WirelessHART network is m=12 [7]. The total utilization of the flow sets is generated from 0.05 to 0.9, with 0.05 as the step length. The range of the period is from $2^{6}$ to $2^{9}$ for every flow in the set [18]. Each flow’s execution time can be calculated by $C_{i}=U_{i}\times T_{i}$. Through Equation (Equation 3), we find that the transmission conflicts caused by $F_{j}$ to $F_{i}$ can be converted into the execution time of flow $F_{j}$ by proportion β, where $\beta =\Delta \left(i,j\right)/C_{j}$. Hence, we assume the upper bound of the transmission conflicts $T^{up}=\beta \times C$. We define α as the cross ratio of a flow to other flows in the set, that is, the number of flows in the collection that have common nodes with that flow in the WirelessHART network.

Under each utilization, we generate 1000 sets of flows and use the average value to represent the characteristics of the flow sets. The experimental parameters are N=100,α=0.1,β=1,δ=1.

### 6.1. Average Value of the Total End-to-End Delay of Flow Sets

The average value of the total end-to-end delay of the flow sets are shown in Figure 3. The delay of the four algorithms increases with the increase in the total utilization of flow sets. We set the average value of total end-to-end delay of sets to zero, when none of the 1000 sets of tasks we generate are available for scheduling. The average value of total end-to-end delay of flow sets becomes zero, when the total utilization exceeds 0.65, by using HLS to assign priority because no set can be scheduled. This means that there is not one set of the 1000 sets we generated that can be scheduled when the utilization exceeds 0.65. In the same way, there is no set that can be scheduled by using WF-D, WF-C, or WF-U when the total utilization exceeds 0.85. At each utilization rate, the delay of the four algorithms is sorted from high to low in the order of HLS, WF-D, WF-U, and WF-C. Among the four algorithms, the optimal algorithm WF-C can reduce the delay by up to 44.18% compared with the worst algorithm HLS when the utilization is 0.6.

### 6.2. Ratio of Schedulable Flow Sets

The ratio of schedulable flow sets refers to the ratio of the scheduled flow sets in the 1000 experimental sets. As shown in Figure 4, the ratio of schedulable flow sets by using the four algorithms decreases from large to small as the total utilization of flow sets increases. When the total utilization of the flow sets is 0.2, the HLS algorithm exhibits unschedulable flow sets, and when the total utilization exceeds 0.6, all flow sets are unschedulable. Similarly, unschedulable flow sets appear when the utilization of WF-D, WF-U, and WF-C algorithms is 0.25, 0.35, and 0.35, respectively. When the utilization rate exceeds 0.85, none of the three algorithms has a schedulable flow set. The WF-C algorithm increases the schedulability ratio by up to 70.7% compared with the HLS algorithm when the utilization is 0.5.

### 6.3. Calculation Time of the Schedulability Analysis

The calculation time of the schedulability analysis represents the time consumed to verify that a flow set is schedulable. The calculation time is shown in Figure 5. We review the WF-D algorithm above. The WF-D only assigns priority according to the deadline of flows in each channel. Different from WF-D, each time WF-C and WF-U assign a priority, they will calculate whether the flow is schedulable under the current priority and choose the highest C and U from all the schedulable flows to assign this priority. In other words, the WF-D algorithm has a pre-allocated priority according to the deadline of the flows and does not need to spend time assigning the priority of the flows. That is why the WF-D algorithm consumes the least amount of time among the four algorithms. As the utilization rate increases, the feasible tree generated by the HLS algorithm increases, and searching for a feasible solution consumes much time. When the utilization exceeds a certain value, due to the high utilization, the number of initial feasible solutions generated by the HLS algorithm becomes smaller, as does the solution space, so the calculation time becomes greatly reduced. Hence, the calculation time of the HLS sharply drops when the utilization exceeds 0.8. The calculation times of the WF-C and WF-U are much shorter than that of the HLS when the utilization is larger than 0.3.

### 6.4. Effect of Other Parameters on the Model

To check the stability of our simulation model and algorithms (WF-C and WF-U), we performed several experiments with changing input parameters. Figure 6a–c show the average value of the total end-to-end delay of the flow sets, the ratio of schedulable flow sets, and the calculation time of the schedulability analysis with N=100,α=0.05,β=1, and δ=1, respectively. The experiments with N=50,α=0.05,β=2, and δ=0.8 are shown in Figure 7.

By observing and comparing the experimental results, we find that the overall trend of the four algorithms remains unchanged even when the input is changed. In other words, the two algorithms we proposed are superior to WF-D and HLS.

Comparative experiments indicate that the four algorithms are similar, even with different input parameters. However, WF-C and WF-U are better than the HLS algorithm.

## 7. Related Work

Priority assignment and schedulability analysis of multi-channel real-time flows in the WirelessHART network are usually treated as problems for multi-processor real-time tasks in a CPU [8]. As a result, we do not know when the level-*k* busy period [9] begins. Therefore, the upper bound of multiprocessor interference is usually calculated instead of the exact solution [19]. Tao et al. proposed a mechanism which can allocate the requirements to user channels based on the different priority levels and ensure that the user with the highest priority will immediately gain channel access [20]. Hossam et al. proposed a new protocol called SS-MAC which can reduce nearly 30% in the worst-case delay [21]. Both Tao and Hossam assume that the deadline of every flow is fixed at 250 ms. Compared with their model, our method, which randomly generates deadlines based on utilization, can be adapted to the system, whose flow deadlines change dynamically. Wei Shen et al. proposed a new scheduling policy called SAS-TDMA to improve the quality of service for the network and reduce the delay for real-time communication [22]. This method improves the reliability of the system in the heavy-noise environment. It is different from our method, which improves reliability by adding redundancy. Furthermore, we use multiple channels to improve the efficiency of the system compared to their method.

The fixed priority model has two global priority assignment strategies, namely, LS and HLS [7]. The LS and HLS algorithms are based on creating a feasible tree of priority assignment strategies and cutting in accordance with the upper and lower bounds of WETED. Several priority assignment strategies are available for multi-processors in the arbitrary deadline model [23,24,25].

Considering the reality, the energy of IoT devices (including wireless sensors) are powered by batteries (IEEE 802.15.4). Therefore, reducing power has a very important value [26]. Weizhe et al. proposed a trusted real-time scheduling model and a successful meta-heuristic method called shuffled frog leaping algorithm (SFLA) [27] to reduce energy consumption. The node residual energy after data flow transmission in linear wireless sensor networks from source nodes and relay nodes was evaluated by Wang et al. [28].

## 8. Conclusions and Future Work

In this paper, we transform the multi-channel TDMA real-time flow model in the WirelessHART network into a real-time task scheduling model in the CPU. On the basis of this transformation, we use the worst-fit method to allocate flows to each channel and provide the calculation formula of the WETED of flows in a single channel. Afterward, we present two heuristic algorithms, WF-C and WF-U, to assign priority. Using the simulation model we built, we find that the two proposed algorithms are more efficient than WF-D and HLS algorithms, and the WF-C algorithm is the most efficient. Compared with HLS, the average value of the total end-to-end delay of WF-C can be reduced by up to 44.18%, and the ratio of schedulable flow sets can be increased by up to 70.7%.

In this paper, we assume that the real-time flows are preemption fixed priority flows. However, there are many kinds of real-time flows such as non-preemption, preemption latency fixed priority flows, and many kinds of dynamic priority real-time flows. The scheduling policy of these real-time flows and the schedulability analysis of these real-time flows can be improved in the future. We intend to propose a new schedulability analysis in the future, so as to reduce the calculation time and reduce the time complexity of schedulability analysis, thus improving the computing efficiency of the system. We also intend to propose a new method of priority assignment for other kinds of real-time flows in the future, so as to improve the ratio of scheduled of sets and thus make the system more stable.

## Figures and Tables

**Figure 1 sensors-18-04242-f001:**
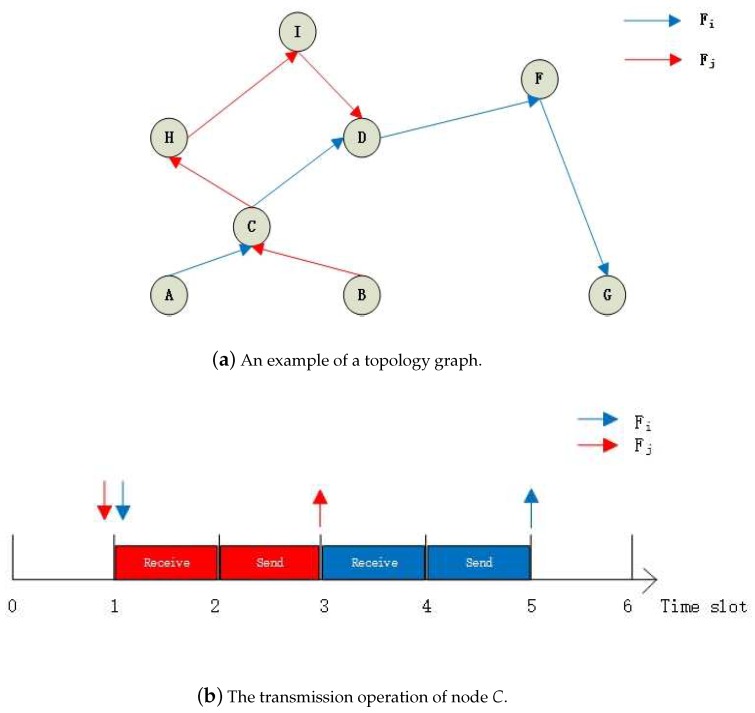
An example of transmission conflicts.

**Figure 2 sensors-18-04242-f002:**
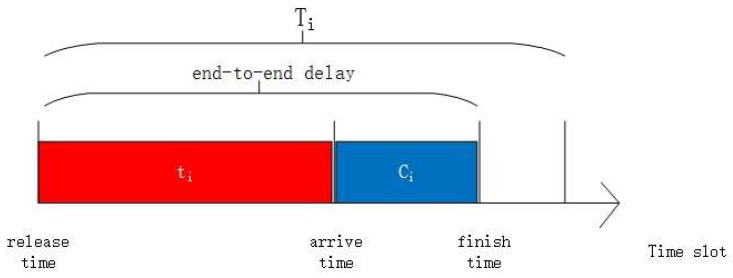
The diagram of the relationship between parameters.

**Figure 3 sensors-18-04242-f003:**
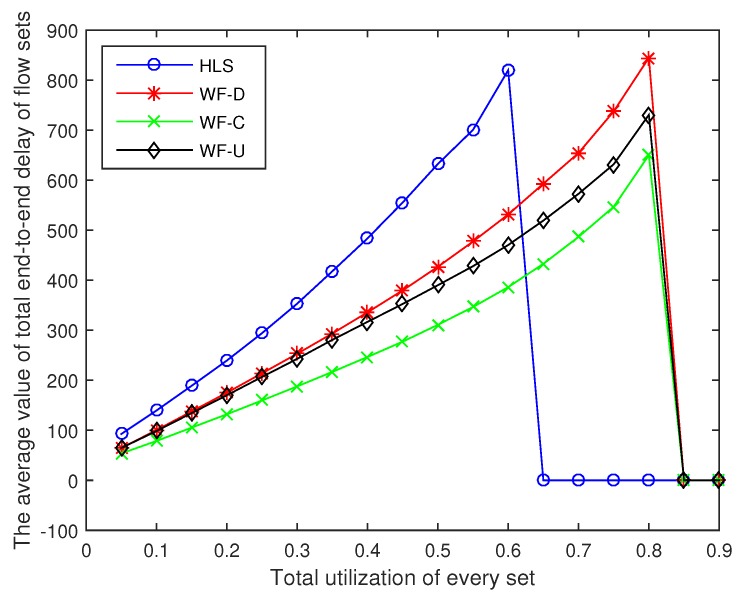
The average value of total end-to-end delay of flow sets.

**Figure 4 sensors-18-04242-f004:**
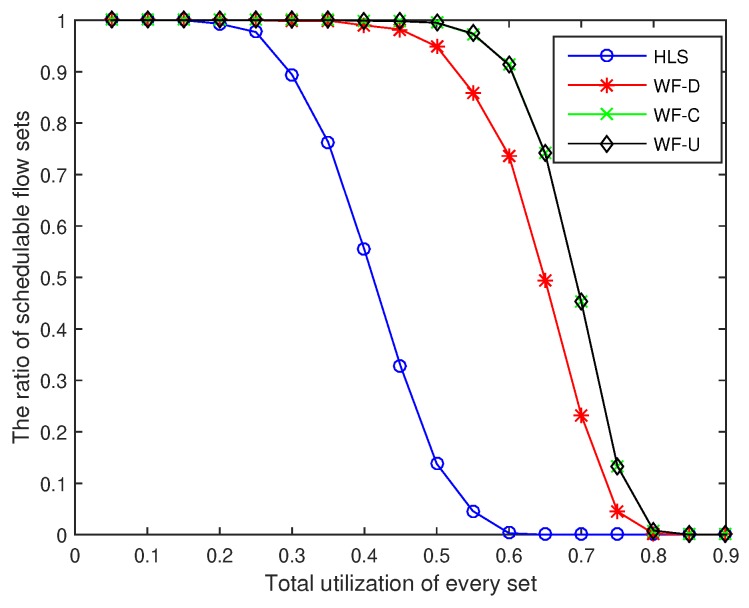
The ratio of scheduled of sets.

**Figure 5 sensors-18-04242-f005:**
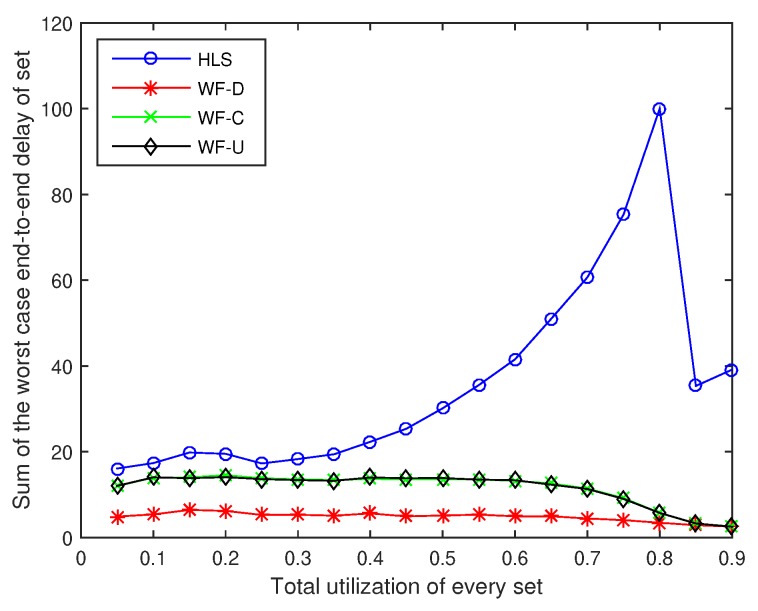
The calculation time of the schedulability analysis.

**Figure 6 sensors-18-04242-f006:**
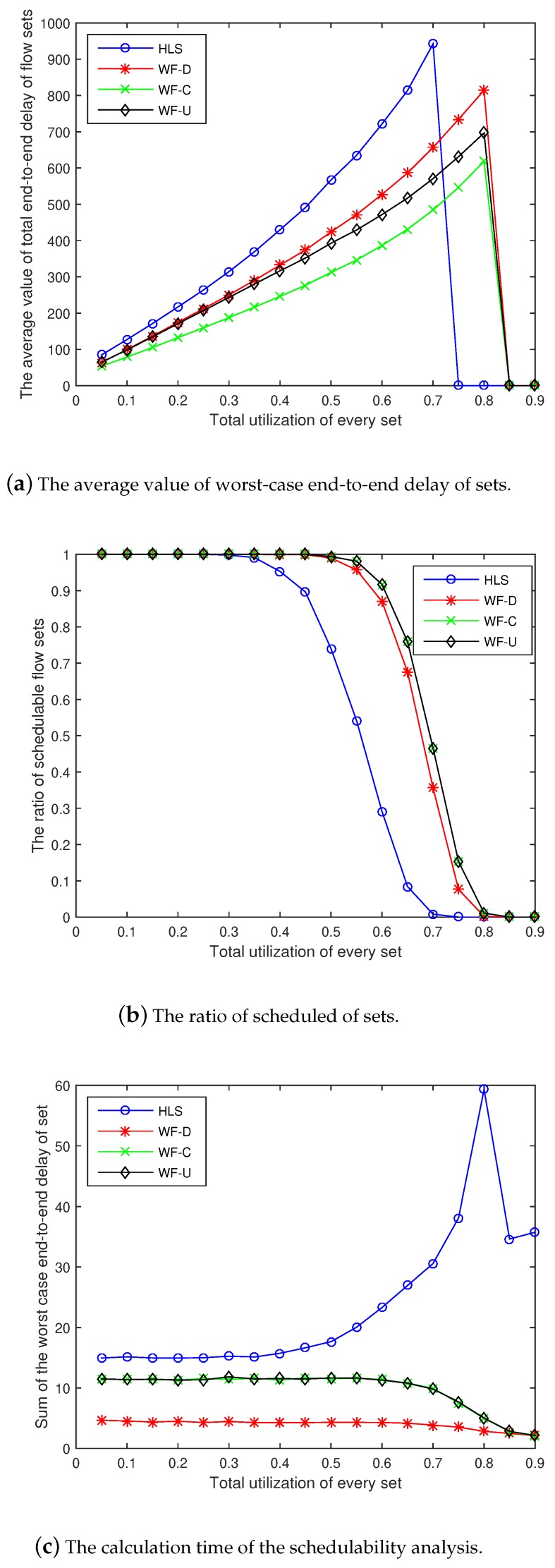
Experiments with N=100,α=0.05,β=1, and δ=1.

**Figure 7 sensors-18-04242-f007:**
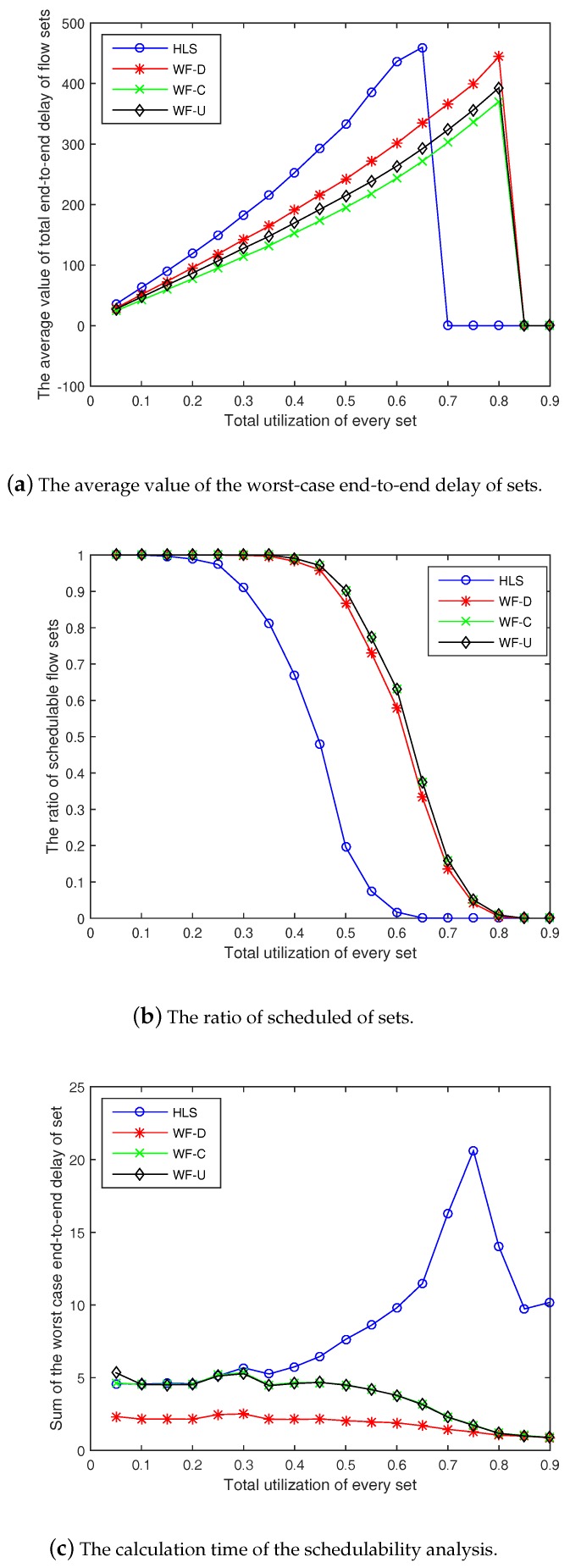
Experiments with N=50,α=0.05,β=2, and δ=0.8.

**Table 1 sensors-18-04242-t001:** The symbols of simulation model.

Symbol	Description
*N*	The number of flows in the set.
*m*	The number of channels.
*α*	The cross ratio of a flow.
*β*	The factor for upper bound of transmission conflicts.
*δ*	The ratio of *D* to *T*.
$T^{up}$	The upper bound of the transmission conflicts.

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
