# Peer review of "A New Method of Priority Assignment for Real-Time Flows in the WirelessHART Network by the TDMA Protocol"

_sensors, 2018, doi:10.3390/s18124242_

Reviewer 1 Report

1. I think that it is difficult for readers to understand transmission conflicts with Figure 1 alone. The authors should present diagrams showing the transmission / reception operation of each node in chronological order. 2. line 206 and Algorithm 3 It seems that these are explanations of WF-U, not WF-C. The authors need to correct these errors. 3. There is no explanation about the HLS algorithm that the authors compare. In order to aid the reader's understanding, the authors should write an overview of the HLS algorithm. 4. As shown in FIG. 2, as the total utilization increases, the average end-to-end delay increases, but there is no explanation that it will become 0 when it exceeds a certain value. The authors should explain the average delay variation shown in Figure 2. 5. The "DM" algorithm indicated by Abstract and introduction is considered to be the worst-fit algorithm described in Algorithm 1, but it is hard to understand because it is not specified. Furthermore, in the graphs after FIG. 2, it is described as "WF-D", and the entire article is not consistent. The authors should modify these descriptions. 6. line 31 TYPO "TMDA"

Author Response

Dear Editors and Reviewer:

Thank you for your letter and for the reviewers’ comments concerning our manuscript entitled “A new method of priority assignment for real-time flows in the WirelessHART network by TDMA protocol (ID: sensors-382082). Those comments are all valuable and very helpful for revising and improving our paper, as well as being important guiding significance to our research. We have studied the comments carefully and have made corrections which we hope will be met with approval. Revised portions are marked with red. The main corrections in the paper and responses to the reviewer’s comments are as following:

Response to Reviewer 1 Comments

Point 1. I think that it is difficult for readers to understand transmission conflicts with Figure 1 alone. The authors should present diagrams showing the transmission / reception operation of each node in chronological order.

Response 1: We have added the descriptions of transmission conflicts in two cases: (1) full transmission conflicts and (2) half transmission conflicts which cost two slots and one slot respectively. Then we described the two cases of transmission / reception operation of representative nodes in chronological order.

We have also changed the original figure1 to figure2a and added two figures (figure2b and figure2c) to explain the transmission / reception operation of representative nodes.

Point 2. line 206 and Algorithm 3 It seems that these are explanations of WF-U, not WF-C. The authors need to correct these errors.

Response 2: We have corrected these errors.

Point 3. There is no explanation about the HLS algorithm that the authors compare. In order to aid the reader's understanding, the authors should write an overview of the HLS algorithm.

Response 3: We have added the overview of the HLS algorithm. Please refer to the references [5] to get more specific descriptions and operations.

Point 4. As shown in FIG. 2, as the total utilization increases, the average end-to-end delay increases, but there is no explanation that it will become 0 when it exceeds a certain value. The authors should explain the average delay variation shown in Figure 2.

Response 4: We have added the explanation that why the average end-to-end delay will become zero when it exceeds a certain value to make Figure 2 clearer.

Point 5. The "DM" algorithm indicated by Abstract and introduction is considered to be the worst-fit algorithm described in Algorithm 1, but it is hard to understand because it is not specified. Furthermore, in the graphs after FIG. 2, it is described as "WF-D", and the entire article is not consistent. The authors should modify these descriptions.

Response 5: We have unified the description of the WF-D algorithm, and we have changed all “DM” into “WF-D”.

Point 6. line 31 TYPO "TMDA".

Response 6: We have corrected this typo.

Reviewer 2 Report

The topic being tackled in this paper is not well de ned and the contribution of the authors is hence not clear.

The text in this paper is confusing which makes the paper hard to follow, especially for Section 1 and 2.

line 17: the listed devices are WirelessHART not IEEE 802.15.4.

line 31: is TDMA suitable for all real-time scenarios?, i.e., is it suitable for acyclic real-time flows?.

Line 39: what is the difference between a real-time flow and a regular flow?

line 74:: how sending the packet in multiple routes can prevent transmission failure in case of blacklisting? I think the authors want to say multiple channels instead of multiple routes.

In section 3: the difference between the defined release time ri and arrival time ai is not clear.

In section 4.2: typically, in WirelessHART networks, the periodic flows are deterministically scheduled (assigned exclusive time slots) for their transmissions to the final destination, even the redundant routes. So, how may a transmission conflict occur in this case?

How the proposed priority strategy schedules flows that have the samepriority?

1, In figures 2, 5 and 6 the average delay drops sharply to zero. Why?

There are a lot of redundant texts in the paper, that is the same information mentioned several times, e.g., line 36/line 64 and line 42/line 79.

There are many typos in the paper, e.g., line 31: TDMA not TMDA,

There is inconsistency in defining the parameters, for instance, in section

2, the authors mention that there are n flow, while in section 3 they mentioned that there are N flows?

3, the authors are suggested to do a proper SOTA since a lot of work for IWSN scheduling is missing. You can refer to some of the following ones:

E. Sisinni, et al. "Industrial Internet of Things: Challenges, Opportunities, and Directions"

D. Yang et al. "An Efficient Retransmission Scheme for Reliable End-to-End Wireless Communication over WSANs"

H. Farag et al "A Delay-Bounded MAC Protocol for Mission- and Time-Critical Applications in Industrial Wireless Sensor Networks"

D. Yang "Safe-WirelessHART: A Novel Framework Enabling Safety-Critical Applications over Industrial WSNs,"

K. Yu et al "Performance Evaluations and Measurements of the REALFLOW Routing Protocol in Wireless Industrial Networks"

F. Dobslaw et al "QoS-Aware Cross-layer Configuration for Industrial Wireless Sensor Networks"

F. Dobslaw et al "End-to-End Reliability-Aware Scheduling for Wireless Sensor Networks"

T. Zhen et al "WirArb: A New MAC Protocol for Time-Critical Industrial Wireless Sensor Networks Applications"

W. Shen et al "SAS-TDMA: A Source Aware Scheduling Algorithm for Real-Time Communication in Industrial Wireless Sensor Networks"

D. Yang et al "CCA-Enabled TDMA Enabling Acyclic Traffic in Industrial Wireless Sensor Networks"

Author Response

Dear Editors and Reviewer:

Thank you for your letter and for the reviewers’ comments concerning our manuscript entitled “A new method of priority assignment for real-time flows in the WirelessHART network by TDMA protocol (ID: sensors-382082). Those comments are all valuable and very helpful for revising and improving our paper, as well as being important guiding significance to our research. We have studied the comments carefully and have made corrections which we hope will be met with approval. Revised portions are marked with red. The main corrections in the paper and responses to the reviewer’s comments are as following:

Response to Reviewer 2 Comments

Point 1. The topic being tackled in this paper is not well dened and the contribution of the authors is hence not clear.

Response 1: We have rewritten Abstract to make the topic and our contributions clearer.

Point 2. The text in this paper is confusing which makes the paper hard to follow, especially for Section 1 and 2.

Response 2: We have added lots of descriptions, examples and figures in Section 1 and 2 to make them clearer and easier to follow.

Point 3. line 17: the listed devices are WirelessHART not IEEE 802.15.4.

Response 3: We have corrected this error.

Point 4. line 31: is TDMA suitable for all real-time scenarios?, i.e., is it suitable for acyclic real-time flows?.

Response 4: We added the description that TDMA suitable for acyclic real-time flows by thinking of them as the real-time flows with infinite length period.

Point 5. Line 39: what is the difference between a real-time flow and a regular flow?

Response 5: We have added the difference between a real-time flow and a regular flow to make it clearer.

Point 6. line 74: how sending the packet in multiple routes can prevent transmission failure in case of blacklisting? I think the authors want to say multiple channels instead of multiple routes.

Response 6: We have rewritten the description of this policy and added an example to make this case clearer.

Point 7. In section 3: the difference between the defined release time ri and arrival time ai is not clear.

Response 7: We added Figure 1 and the descriptions of ri and ai to make the difference between them clearer.

Point 8. In section 4.2: typically, in WirelessHART networks, the periodic flows are deterministically scheduled (assigned exclusive time slots) for their transmissions to the final destination, even the redundant routes. So, how may a transmission conflict occur in this case?

Response 8: We have rewritten the description of the transmission conflict make it clearer . We added two examples of the transmission conflict and two figures in Section 4.2 to explain a transmission conflict occur.

Point 9. How the proposed priority strategy schedules flows that have the same priority?

Response 9: We added the description of priority assignment to explain there are not two flows have the same priority.

Point 10. In figures 2, 5 and 6 the average delay drops sharply to zero. Why?

Response 10: We have added the explanation that why the average end-to-end delay will become zero when it exceeds a certain value.

Point 11. There are a lot of redundant texts in the paper, that is the same information mentioned several times, e.g., line 36/line 64 and line 42/line 79.

Response 10: We have rewritten these information to avoid of repetition.

Point 12. There are many typos in the paper, e.g., line 31: TDMA not TMDA,

Response 12: We have corrected these typos.

Point 13. There is inconsistency in defining the parameters, for instance, in section 2, the authors mention that there are n flow, while in section 3 they mentioned that there are N flows?

Response 13: We have unified the definition of parameters.

Point 14. the authors are suggested to do a proper SOTA since a lot of work for IWSN scheduling is missing. you can refer to some of the following ones:

E. Sisinni, et al. "Industrial Internet of Things: Challenges, Opportunities, and Directions"

D. Yang et al. "An Efficient Retransmission Scheme for Reliable End-to-End Wireless Communication over WSANs"

H. Farag et al "A Delay-Bounded MAC Protocol for Mission- and Time-Critical Applications in Industrial Wireless Sensor Networks"

D. Yang "Safe-WirelessHART: A Novel Framework Enabling Safety-Critical Applications over Industrial WSNs,"

K. Yu et al "Performance Evaluations and Measurements of the REALFLOW Routing Protocol in Wireless Industrial Networks"

F. Dobslaw et al "QoS-Aware Cross-layer Configuration for Industrial Wireless Sensor Networks"

F. Dobslaw et al "End-to-End Reliability-Aware Scheduling for Wireless Sensor Networks"

T. Zhen et al "WirArb: A New MAC Protocol for Time-Critical Industrial Wireless Sensor Networks Applications"

W. Shen et al "SAS-TDMA: A Source Aware Scheduling Algorithm for Real-Time Communication in Industrial Wireless Sensor Networks"

D. Yang et al "CCA-Enabled TDMA Enabling Acyclic Traffic in Industrial Wireless Sensor Networks"

Response 14: We did some research on IWSN scheduling and added some of these references into our manuscript.

Reviewer 3 Report

This paper proposes the worst-fit method to allocate real-time flows to each channel in the WirelessHART network based on the multi-channel TDMA protocol, and proved that the priority assignment in a single channel is NP-Complete, and then presented two heuristic algorithms, WF-C (longest the execution time) and WF-U (highest utilization), to assign priority.

The TDMA  protocol and the end-to-end delay analysis are clearly presented and the problem formulation are well-defined. The experiments with a new simulation model based on the reality demonstrate the proposed WF-C and WF-U algorithms outperformed HLS and WF-D.

However, many descriptions are over simplified and the authors also did not cite the related papers. More discussions are needed to distinguish the proposed work with the existing works. It’s difficult for non-expert readers to follow. For example, In Section 5, ‘’We distribute the flows to the channels. Then, multi-channel priority assignment can be regarded as several single-channel priority assignment problems.” The author should explain more clearly. In Section 7, “Considering the power requirement in wireless sensor networks...” Cannot figure out why mentioned the power requirement.

Also, the assumption is a little simple. This paper fails to convince me that the method can be applied in real world. In our opinion, the paper is more suitable for submitting a conference. For example, can point out why use the worst-fit algorithm to allocate the flows. Or discuss the results of the different algorithms results or consider more complicated problem, such as multi-hop or single hop networks. Or discuss time complexity.

Section 5.2 and Section 5.3 should be merged, the contexts are almost the same.

Some notations are difficult to follow, e.g., Rek, Re, ReF, what is Rj?

Typo: Intothe, the title and function of Algorithm 3 are incorrect, wherek, ACK, UCH, CH, she is a girls

Author Response

Dear Editors and Reviewer:

Thank you for your letter and for the reviewers’ comments concerning our manuscript entitled “A new method of priority assignment for real-time flows in the WirelessHART network by TDMA protocol (ID: sensors-382082). Those comments are all valuable and very helpful for revising and improving our paper, as well as being important guiding significance to our research. We have studied the comments carefully and have made corrections which we hope will be met with approval. Revised portions are marked with red. The main corrections in the paper and responses to the reviewer’s comments are as following:

Response to Reviewer 3 Comments

Point 1. However, many descriptions are over simplified and the authors also did not cite the related papers. More discussions are needed to distinguish the proposed work with the existing works. It’s difficult for non-expert readers to follow. For example, In Section 5, ‘’We distribute the flows to the channels. Then, multi-channel priority assignment can be regarded as several single-channel priority assignment problems.” The author should explain more clearly. In Section 7, “Considering the power requirement in wireless sensor networks...” Cannot figure out why mentioned the power requirement.

Response 1: We have rewritten the description of multiple channels mapping in Section 5 to make it clearer and easier for readers to follow. And we have distinguished the Saifullah et al. work with our contribution.

We have rewritten the description in Section 7 to explain why mentioned the power requirement.

Point 2. Also, the assumption is a little simple. This paper fails to convince me that the method can be applied in real world. In our opinion, the paper is more suitable for submitting a conference. For example, can point out why use the worst-fit algorithm to allocate the flows. Or discuss the results of the different algorithms results or consider more complicated problem, such as multi-hop or single hop networks. Or discuss time complexity.

Response 2: We added three methods for allocating the flows to the channels. We also added the descriptions and characters of them to explain why we use the worst-fit method to allocate the flows in Section 5.

We added Section 5.3 to discuss the time complexity of WF-C and WF-U algorithms.

Point 3. Section 5.2 and Section 5.3 should be merged, the contexts are almost the same.

Response 3: We have merged Section 5.2 and Section 5.3 and rewritten the descriptions of WF-U algorithm to make it clearer.

Point 4. Some notations are difficult to follow, e.g., Rek, Re, ReF, what is Rj?

Response 4: We have removed the Re and rewritten the description of Rek, UChk, AChk to make readers easier to follow. We added the description of ReF to make it clearer. Rj means the worst case end-to-end delay of flow Fj. You can refer to Equation 3 in Section 5 to get the definition of Rj.

Point 5. Typo: Intothe, the title and function of Algorithm 3 are incorrect, where k, ACK, UCH, CH, she is a girls.

Response 5: We have corrected these typos.

Round  2

Reviewer 1 Report

For the comments presented by reviewers, I confirmed that the authors politely corrected the revisions.

Author Response

Dear Editors and Reviewer:

Thank you for your letter and for the reviewers’ comments concerning our manuscript entitled “A new method of priority assignment for real-time flows in the WirelessHART network by TDMA protocol (ID: sensors-382082). We are honored to receive the comment from you. Thank you for taking the time out of your busy schedule to review our manuscript.

Reviewer 2 Report

The authors included a few new related work but it would be interested if the authors could elaborate more about related work and also compare with 3-4 related work. 

Author Response

Dear Editors and Reviewer: Thank you for your letter and for the reviewers’ comments concerning our manuscript entitled “A new method of priority assignment for real-time flows in the WirelessHART network by TDMA protocol” (ID: sensors-382082). Those comments are all valuable and very helpful for revising and improving our paper, as well as being important guiding significance to our research. We have studied the comments carefully and have made corrections which we hope will be met with approval. Revised portions are marked with red. The main corrections in the paper and responses to the reviewer’s comments are as following: Response to Reviewer 2 Comments Point 1. The authors included a few new related work but it would be interested if the authors could elaborate more about related work and also compare with 3-4 related work. Response 1: We added some descriptions of these related works to make them clearer and compared them with our method.

Reviewer 3 Report

The paper has improved a lot from the original submission. It is pleasing to see the authors have endeavored to address the reviewers' comments.

Some further suggestions for improving the paper are given below.

In Figure 5 and Figure 7(c), please explain HLS sharply dropped when the calculation time of the schedulability analysis us from 0.8 to 0.85. Also, according to the result for the calculation time of the schedulability analysis, WF-D seems better than WF-C and WF-U. Please also explain the reason in the experiment.

Please also add future work, e.g., addressing future problems, how to extend your work, etc.

The figures number should be appeared in order, e.g., Figure 1 and Figure 2.

The article still has many typos, and could be improved further, e.g, re ceive, and\alpha, 1000, he comparative.

Author Response

Dear Editors and Reviewer:

Thank you for your letter and for the reviewers’ comments concerning our manuscript entitled “A new method of priority assignment for real-time flows in the WirelessHART network by TDMA protocol (ID: sensors-382082). Those comments are all valuable and very helpful for revising and improving our paper, as well as being important guiding significance to our research. We have studied the comments carefully and have made corrections which we hope will be met with approval. Revised portions are marked with red. The main corrections in the paper and responses to the reviewer’s comments are as following:

Response to Reviewer 3 Comments

Point 1. In Figure 5 and Figure 7(c), please explain HLS sharply dropped when the calculation time of the schedulability analysis us from 0.8 to 0.85. Also, according to the result for the calculation time of the schedulability analysis, WF-D seems better than WF-C and WF-U. Please also explain the reason in the experiment.

Response 1: We have added the explanation of HLS sharply dropped when the calculation time exceeds 0.8. And we added the explanation of the calculation time of WF-D is less than the calculation time of WF-C and WF-U.

Point 2. Please also add future work, e.g., addressing future problems, how to extend your work, etc. The figures number should be appeared in order, e.g., Figure 1 and Figure 2.

Response 2: We added future work, including future problems and how to extend our work.

We have changed the order of the two pictures to make them in order.

Point 3. The article still has many typos, and could be improved further, e.g, re ceive, and\alpha, 1000, he comparative.

Response 3: We have corrected these typos.
